environmental science/computational biology

high alpine grazing, transhumance, climate change, lactation curve modelling, Braunvieh dairy cattle, environmental variables

**Author for correspondence:**
Stéphane Joost
e-mail: stephane.joost@epfl.ch

[†]This author should be considered as co-senior author.

# Big dairy data to unravel effects of environmental, physiological and morphological factors on milk production of mountain-pastured Braunvieh cows

Solange Duruz[1], Elia Vajana[1], Alexander Burren[2], Christine Flury[2,†] and Stéphane Joost[1]

[1]Laboratory of Geographic Information Systems (LASIG), School of Architecture, Civil and Environmental Engineering (ENAC), Ecole Polytechnique Fédérale de Lausanne (EPFL), Lausanne, Switzerland
[2]School of Agricultural, Forest and Food Sciences, Bern University of Applied Sciences, Zollikofen, Switzerland

 SD, 0000-0002-1235-6747; SJ, 0000-0002-1184-7501

The transhumance system, which consists in moving animals to high mountain pastures during summer, plays a considerable role in preserving both local biodiversity and traditions, as well as protecting against natural hazard. In cows, particularly, milk production is observed to decline as a response to food shortage and climatic stress, leading to atypical lactation curves that are barely described by current lactation models. Here, we relied on 5 million monthly milk records from over 200 000 Braunvieh and Original Braunvieh cows to devise a new model accounting for transhumance, and test the influence of environmental, physiological and morphological factors on cattle productivity. Counter to expectations, environmental conditions in the mountain showed a globally limited impact on milk production during transhumance, with cows in favourable conditions producing only 10% more compared with cows living in detrimental conditions, and with precipitation in spring and altitude revealing to be the most production-affecting variables. Conversely, physiological factors such as lactation number and pregnancy stage presented an important impact over the whole lactation cycle with 20% difference in milk production, and alter the way animals respond to transhumance. Finally, the considered morphological factors (cow height and foot angle) presented a smaller impact during the whole lactation

cycle (10% difference in milk production). The present findings help to anticipate the effect of climate change and to identify problematic environmental conditions by comparing their impact with the effect of factors that are known to influence lactation.

# 1. Introduction

Transhumance, which consists of moving livestock to high mountain pastures in the summer months, provides both ecological and socio-cultural services to the human populations living in the mountainous regions of many European countries [1–3]. Indeed, transhumance-annexed grazing sustains and preserves endemic plant communities [4], feeds local cattle to produce traditional alpine cheese and attracts many tourism-related activities [5]. Further, it counteracts land abandonment in mountain areas and, therefore, contributes towards preserving landscape against scrub growth and vegetation encroachment [6], as well as natural hazards such as avalanches [7] and wild fires [5]. The term 'alping' (a translation of the German word 'Alpung' or its French equivalent 'montée à alpage') will be used here to describe the approximately 100 days that dairy cattle spend on alpine pastures during the summer months. Similarly, animals brought to mountain pastures will be referred to as 'alped' cows, and the alpine summer pastures will be called 'alps'.

Despite such ecological and social benefits, the surface dedicated to alping decreases each year (approx. 2400 ha yr$^{-1}$ [8]), and a questionnaire-based study revealed in 2010 that one-third of the participating breeders intend to probably abandon the transhumance practice in the following decades. In summer 2018, 107 000 dairy cows were alped in Switzerland for approximately 100 days [9]. A steep drop in milk production is observed during this period, which hampered the evaluation of lactation curves through standard models that assume a linear decrease in production [10] after the maximum milk yield is reached (i.e. approx. 100 days after calving) [11]. Among the explanations proposed to interpret such a detrimental effect on productivity are the feed deficit intake due to the meagre grassland as found in high alpine pastures, as well as the need to tackle environmental stress due to new and sometimes harsh habitat conditions [12]. On the other hand, milk composition is known to change during alping [13,14] and results in the production of highly valuable milk products such as butter and alp cheese.

Milk production and quality is notoriously affected by a wide variety of environmental factors, including calving season and vegetation types composing animals' diet [15–18]. Environmental temperature is also known to directly affect cattle productivity because of heat [17] or cold [19] stress. Furthermore, milk quality and production of alped cows are expected to be indirectly affected by global warming, as forage quality and biomass productivity of alpine sites are likely to decrease with increasing temperature and decreasing precipitation [20,21].

Despite the existence of huge databases storing monthly milk records for several European cattle breeds, no effort has been produced so far (at least to our knowledge) to exploit such information and understand the ways alping affects milk productivity [22]. Indeed, most of the existing literature focuses on small experiments (with sample size less than 100) mainly restricted to comparing two groups of animals in different environmental conditions, so as to investigate the potential effects of altitude [23], vegetation type [23,24], supplemental feeding [25,26], calving season [27] or breed [12,27,28]. Furthermore, no adaptation of general models of lactation curves [11] have been proposed to account for alping, which hinders a straightforward comparison of lactation curves for alped cows. Last but not least, the overall impact of environmental factors and global warming on milk production during alping is also still unknown.

For these reasons, a better understanding and characterization of the impacts of transhumance on milk production and the way production is influenced by environmental factors is needed. To fill this gap, we relied on over 5 million monthly test-day milk records collected between 2000 and 2015 from more than 200 000 Braunvieh cows, a local Swiss cattle breed well adapted to the alpine pastures. Then, we used this information to: (i) devise a new mathematical model to fit lactation during alping, and (ii) investigate the influence of the environment on milk production during alping, and compare it with the effect of physiological and morphological factors. This can be achieved thanks to biogeoinformatics which takes advantage of georeferenced animal data in order to link biological and environmental information with the help of advanced informatics tools [29].

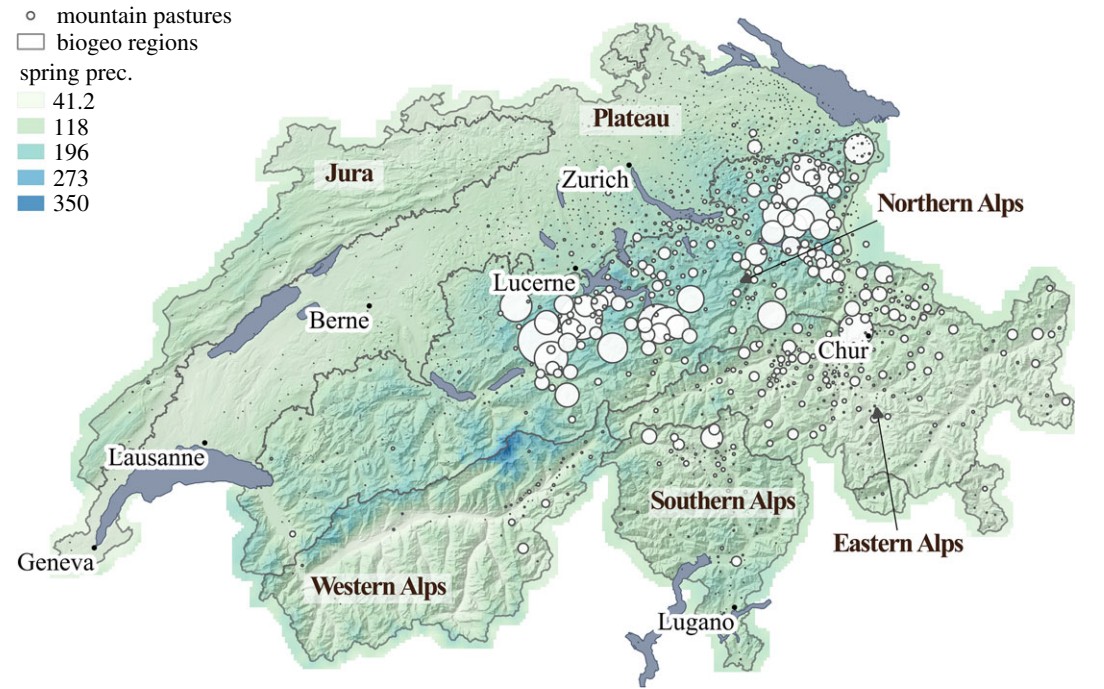

**Figure 1.** Geographical location of the alps hosting Braunvieh cows (white circles), with average monthly precipitation in millimetre between April and July 2015 in the background (chosen as example year). Frontiers of biogeographic regions are also reported. The size of the circles is proportional to the number of milk records taken at a given alp, the biggest and smallest circles encompassing 28 923 and 1 records, respectively. The majority of the alps hosting Braunvieh cows are located in Northern and in the Eastern Alps biogeographic regions.

## 2. Data

### 2.1. Milk records and animal information

Milk records from all alped Braunvieh cows were provided for the period 2000–2015 by the Braunvieh Schweiz AG breeding association. Importantly, a direct comparison with non-alped cows was not possible because we did not have access to these data. However, as milk measurements of alped cows entail records from both the lowland farm and the alp, the estimation of milk production in both situations was feasible. The full dataset is composed of 5 681 498 test-day records (methods A4 and AT4 according to ICAR guidelines [30]), including 616 081 lactations derived from a total of 245 313 cows. In line with national and international rules, milk records are taken approximately on a monthly basis, with the first record taken between the 5th and 42nd day after calving. Each test-day record includes information on the following traits: milk (kg), fat (kg and %), protein (kg and %), somatic cell count (1000 cells ml$^{-1}$). To keep the reader focused on the main thread of the article, our study specifically analyses milk production in terms of quantity (milk yield); however, results from computations with protein and fat content and yield are also available in electronic supplementary material, S2–S5. Out of the total number of records, 1 481 387 were taken in the alps, whose altitudes were systematically stored in the database, while their precise locations were documented in 95% of the cases (figure 1). The first record in the alp is usually taken within the first 4 days after arrival, and is followed by three more records in the alp to encompass the entire alping period (typically 100 days). Moreover, to morphologically describe animals, linear type description and classification of cows are scored during the first lactation of all cows of the database. In our study, we considered the body height at withers and the scores (1–9) for foot angle. In addition, insemination data for each lactation (date, sire's name) are also available.

A stringent data quality control procedure was applied prior to analysis to remove: (i) incomplete years (which resulted in removing beginning of 2000 as well as end of 2015 due to missing lactation records); (ii) cows with average interval between first and last insemination longer than 100 days (as computed over the first three lactations); (iii) cows that had their first calf while being younger than 2 years, or older than 4 years; (iv) cows belonging to breeds different from the Braunvieh or Original Braunvieh; (v) cows with parents other than Braunvieh or Original Braunvieh; (vi) lactations shorter than 270 days; (vii) lactations with calving interval shorter than 290 days; (viii) lactations with alps

below 1100 m.a.s.l. or above 2600 m.a.s.l.; (ix) lactations with calving happening between March and August; (x) lactations from cows that had already calved more than nine times; (xi) lactations with the first record taken after the 42nd day after calving; (xii) lactations with records taken before calving; (xiii) records taken before the 5th day and after the 500th day after calving; and (xiv) the second alping season (i.e. final part of lactation curves) from animals that are alped twice in the same lactation. After filtering, we obtained a final dataset composed of 3 527 138 records over 371 696 lactations from 175 474 cows.

## 2.2. Factors influencing milk characteristics

Milk characteristics are known to be influenced by different factors. Meaningful predictor variables were then selected according to literature review, by assuming the same factors to be relevant in both lowland and mountain conditions. As a result, climatic and environmental indices [19,31] were taken into account together with physiological (lactation number, pregnancy stage [32,33]) and morphological factors (table 1).

## 2.3. Climatic data

Climate has been observed to influence milk production [34]. Consequently, maximum and mean temperature [23] as well as daily rainfall [24] were extracted from the meteoswiss Grid-Data products database. This dataset is derived by interpolation of records from several weather stations across Switzerland, and consists of 2 km resolution raster files (1 km resolution from the year 2014 and on). Further, daily average wind speed and relative humidity were obtained from, respectively, 440 and 495 meteoswiss weather stations. We then interpolated these values between stations to obtain a continuous representation of the variables, with a squared inverse-distance weighting (IDW) [35] within a maximum distance of 50 km.

On the basis of such environmental data, the temperature–humidity index (THI) and cold stress index (CSI) were computed following Bryant *et al.* [19]. These indices assess all relevant climatic conditions for the evaluation of 'hot'/'cold' sensation instead of focusing on temperature only:

$$THI = 0.8T + \left(\frac{RH}{100} \cdot (T - 14.4)\right) + 46.4, \tag{2.1}$$

with $T$ being the maximum daily temperature (°C) and $RH$ the relative humidity (%), and

$$CSI = (11.7 + (3.1 \cdot WS^{0.5})) \cdot (40 - T) + 481 + 418 \cdot (1 - e^{-0.04 \cdot rain}), \tag{2.2}$$

with $WS$ being the daily mean wind speed (m s$^{-1}$), $T$ the mean daily temperature (°C) and *rain* the daily precipitation (mm). These indices were computed over a 3- and 30-day period to account for short/long heat waves/cold spells, respectively.

## 2.4. Digital elevation model

Due to the coarse spatial resolution of temperature data (2 km), a correction of −0.45°C/100 m (i.e. the observed temperature gradient in the dataset) was applied to account for local variation in temperature due to topography. This correction was achieved using both the digital elevation model (DEM) DHM25 dataset produced by swisstopo [36] and the recorded altitude of the alp available in the dataset. The digital model DHM25 is a three-dimensional representation of the earth's surface in Switzerland, as based on the elevation data from the Swiss National Map 1:25 000 (NM25). A symmetric 25 m grid matrix model is then interpolated starting from the digitized contour lines and spot heights from NM25. Comparisons among control points show an average accuracy of the produced model of 2–3 m for the pre-Alps and Alps, respectively.

## 2.5. Biogeographic region

The Federal Office for Environment (FOEN) divided Switzerland into six biogeographic regions [37], obtained using fauna and flora data and aggregating areas with common species. Species distributions being strongly related to the relief, these regions reflect in fact the topography of the country. Most of the alps hosting Braunvieh cows appear to be located in the Northern and Eastern Alps biogeographic regions. More rainfall occurs in the Northern Alps when compared with the eastern side (figure 1), because the mountain chain acts as a barrier to precipitation coming from the west and north [38].

**Table 1.** List of factors included in the present study with supposed influence on lactation during alping. Factor-specific cut-off values are reported in the last column. These values are used to assess factor-specific effects on lactation (see Methods for an exhaustive explanation).

| | name | description | group cut-off |
| --- | --- | --- | --- |
| environmental | temperature humidity index (3 days) | climatic index based on temperature and humidity, averaged over 3 days before milk record at the alp. See section on climatic data | 59.4–65.4/>65.4 |
| | temperature humidity index (30 days) | climatic index based on temperature and humidity, averaged over 30 days before milk record at the alp. See section on climatic data | 59.6–63.1/>63.1 |
| | cold stress index (3 days) | climatic index based on temperature, wind speed and precipitation, averaged over 3 days before milk record at the alp. See section on climatic data | 960.1–1045.9/>1045.9 |
| | cold stress index (30 days) | climatic index based on temperature, wind speed and precipitation, averaged over 30 days before milk record at the alp. See section on climatic data | 997–1042.9/>1042.9 |
| | spring precipitation | average monthly precipitation (mm) between April and July; computed for each year, at the location of each alp | <120.6/>155.3 |
| | biogeographic region | only regions with sufficient sample size were retained, and, therefore, two categorical variables were created. See section on biogeographic regions. | North Alp/East Alp |
| | altitude | altitude (m) of the highest alp during the lactation cycle | <1600/>1900 |
| | altitude difference | difference in altitude between the highest alp and the lowland farm. | <641/>1021 |
| | aspect 100 m | aspect of the alp (North/South facing) as based on 100 m resolution DEM | 300–60/120–240 |
| | aspect 1 km | aspect of the alp (North/South facing) as based on 1 km resolution DEM | 300–60/120–240 |
| physiological | lactation number | number of lactations the cow experienced since birth (correlated with animal age) | 1st lact/≥3rd lact |
| | pregnancy stage | pregnancy stage (days) at the beginning of alping | <73/>153 days |
| morphological | height of animal | height at withers (cm) | <139/>143 |
| | foot angle | a score between 1 and 9 (with 9 being the steepest) | <4/>6 |

# 3. Methods

## 3.1. Lactation curve modelling

A lactation curve is usually estimated from one single cow with repeated observations along a lactation cycle and with records taken on a daily/weekly basis [11]. Here, test-day milk records were collected monthly, making the individual-based estimates of lactation impossible because of the over-parametrization issue faced when the number of observations is small (typically 10 monthly measurements during a whole cycle) with regard to the number of parameters to estimate, particularly when describing a complex curve like the one of alped cows (six parameters, see below). Moreover, a measurement is highly influenced by local temporal variations linked to some momentary discomfort of the animal, so that the curve resulting from monthly records is exceedingly noisy. Therefore, we analysed averaged values by computing, for each test-day, the mean of all available records of that particular day in milk (DIM, or number of days after calving). Given that records from the same animal are one month apart but that they are not taken on the same DIM for all animals, the average of milk records for each test-day will constitute a smooth curve with daily values (as displayed in point observations of figure 2). As dates at which cows are alped or brought back to the lowland farm slightly differ among animals, records from cows remaining at the lowland farm during the alping season (between 15 May and 31 August) were excluded from this average computation, while only cows at the lowland farm were considered in the average outside this time frame. Moreover, cows were grouped according to their calving month. Finally, when fitting the curve, each averaged milk yield was weighted according to the number of observations on that day.

Several models have been proposed to describe lactation curves [39], with the Wood, Wilmink, Ali-Schaeffer (AS) and Legendre polynomial formulations being the most popular [40]. Among these mathematical formulations, Wilmink proposes a linear equation that is retained in the present work, given its inherent simplicity and good performance [40]. This model is written as

$$Y_t = a + b \cdot e^{-k \cdot t} + c \cdot t, \tag{3.1}$$

where $Y_t$ is the observed variable (milk yield), $t$ is the DIM and $a$, $b$, $c$ and $k$ are the parameters to estimate. However, $k$ is usually set to 0.1 to make this equation linear [40]. To validate this value with our data, nonlinear regressions were also run with six test curves (one for each calving month) and the obtained values for $k$ were between 0.05 and 0.37.

Here, we introduce additional terms to equation (3.1) in order to explicitly account for the transhumance effect. Particularly, alping has been observed to severely affect milk production, with alped animals showing a steeper linear decrease than before alping (figure 2). Further, alped cows usually experience a small yet rapid boost shortly after their return to the lowland farm, followed by a softer decline in milk production. Taking these observations into account, we then propose to adapt equation (3.1) as follows:

$$Y_t = a + b \cdot e^{-k \cdot t} + c \cdot t + d \cdot \max(0, t - t_1) + f \cdot \max(0, \text{ceiling}(t - t_2)/305) + g \cdot \max(0, t - t_2). \tag{3.2}$$

Where $t_1$ is the DIM at which the cow is alped, and $t_2$ is the DIM at which the cow is brought back to the lowland farm. Importantly, the expression $d \cdot \max(0, t - t_1)$ is the expected linear decrease during alping, so that the $d$-parameter reflects the effect of alping. The $f \cdot \max(0, \text{ceiling}(t - t_2)/305$ captures the expected boost in production after alping and $g \cdot \max(0, t - t_2)$ represents the linear decrease in milk yield after alping; in the latter arguments, the max() term ensures the model to be only affected during and after alping, respectively, while the ceiling expression (i.e. round to the upper integer) constructs a binary operator (0/1) to recreate the instantaneous boost after the return to the lowland farm. In our case, $t_1$ and $t_2$ were determined independently for each calving month. The proposed equation only works for a standard lactation period of 305 days.

The $d$-parameter enables the estimation of the loss in milk yield associated with alping over a given period of time. Indeed, the amount of milk lost during alping for a period of $x$ days can be approximated with

$$Y_{\text{loss}} = \frac{d \cdot x^2}{2}. \tag{3.3}$$

However, it is essential that the model fits well the beginning of the curve for this equation to work, which can be achieved by artificially increasing the weight of point measurements before the transhumance. Thus, weights before alping were multiplied by 100 when investigating the $d$-parameter depending on the

calving month (figures 2 and 3). Furthermore, as older cows tend to calve later in the season, thereby creating a correlation between lactation number and calving month, the impact of alping according to the calving month is entangled with lactation number. Therefore, when examining milk production and the impact of alping for each calving month, only cows in their first lactation were considered (figure 3).

Ordinary linear regression models were then computed in R using the lm() function of the stats package [41] to estimate parameters in equation (3.2).

## 3.2. Measuring the effect of influencing factors

For the sake of interpretation, all influencing factors (i.e. explanatory variables) were grouped into environmental, physiological and morphological categories (table 1). The effect of influencing factors was tested by comparing milk records produced in conditions as dissimilar as possible. Importantly, since the low number of measurements per animal imposed the use of averages, effect determination was not possible through classical regression models. Consequently, groups were created according to the first and third tertile of the distributions, in order to include animals from the most contrasted situations (environmental, physiological and morphological) while retaining enough observations to guarantee a sufficient statistical power. Since productivity is known to be optimized with mild weather conditions [34], exceptions were made for THI and CSI where the second and the third tertiles were used as the two contrast groups instead of the first and third tertile.

Group membership was assessed through the creation of a dummy variable assuming the value of 1 if belonging to the group considered, 0 otherwise. Then, the impact of influencing factors was computed by adding an interaction term to equation (3.2) that allows chosen parameters to vary as a function of the group. The here-defined environmental variables affect milk production during the alping stay only. Accordingly, lactation curves were modelled only until the end of the alping season (meaning the $f$ and $g$ parameters not to be estimated), with the sole $d$-parameter varying as a function of the group. By contrast, physiological and morphological factors influence the whole lactation cycle, so that all terms of equation (3.2) (coefficients $a$, $b$, $c$, $d$, $f$ and $g$) are allowed to vary as a function of the group.

Within-group production was estimated both at the lowland farm and in the alps for physiological and morphological factors or during alping season only for environmental factors, by integrating the area under the lactation curve. The between-group difference was then assessed by computing the percentage of the difference in milk production with respect to the reference group, this group being arbitrarily chosen as the one with the highest milk production during alping. The difference in the $d$-parameter ($\Delta d$) between the two groups is then also displayed to show how differently the concerned groups were impacted by alping. As the response differs according to the calving months, results were computed for each calving month separately and the months of September and February were chosen as representative of autumn and winter calving, respectively.

## 3.3. Significance testing

Log-likelihood ratio tests were performed to investigate both the impact of adding the parameters $d$, $f$ and $g$ to the Wilmink model, and of the considered influencing factors. When testing the addition of parameters $d$, $f$ and $g$ to the Wilmink equation, equations (3.1) and (3.2) were considered as null and alternative models, respectively; when testing the influencing factors, the null model was constructed by removing the interaction between the dummy variable group and the parameters of equation (3.2).

The resulting $G$-score test-statistics were then converted into $p$-values, which were further corrected for multiple testing by means of Bonferroni's approach [42]. $G$-scores are efficient ways of testing the performance of a nested model, and are slightly less conservative than Wald scores [43]. This seemed appropriate here since the applied correction for multiple testing is already sufficiently conservative. $G$-scores were evaluated using the lrtest() function from the lmtest R-package [44].

# 4. Results

## 4.1. Lactation curve modelling

Overall, the proposed equation fits both the drop in milk production due to alping and the tail of the lactation curve, as illustrated here for the calving months of September and February (figure 2). In particular, the terms added to the Wilmink equation (equation (3.2)) significantly increase the full

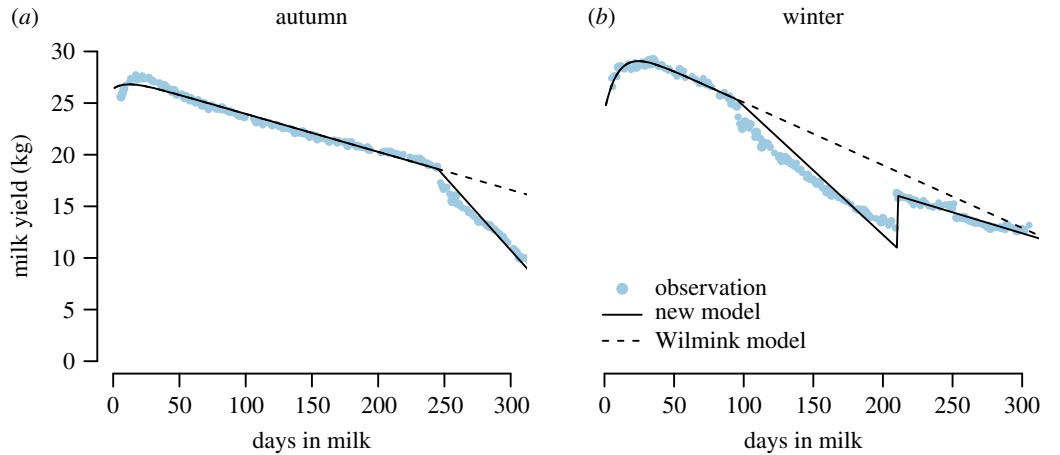

**Figure 2.** Lactation curves as derived from the proposed model (full line) and the Wilmink model (dashed line) for cows that calved in September (*a*) and February (*b*). The Wilmink model was fitted using points from the beginning of the curve only, i.e. before alping. Each dot represents the average of milk records per day. When $t > 245$ (*a*) and between 95 and 210 (*b*), records from the alp only are used to calculate the average, while records from the lowland farm only are included for the remaining time frame.

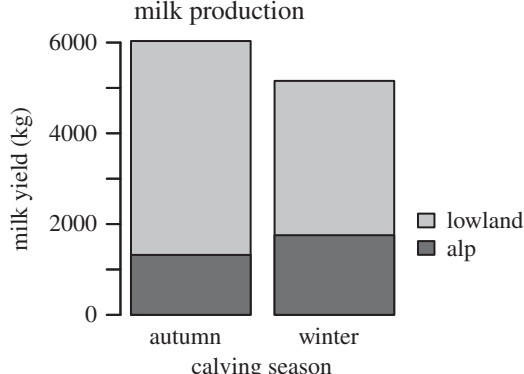

**Figure 3.** Milk production during alping (black) and from the lowland farm (grey) is reported for autumn and winter calving, as represented by the months of September and February, respectively. Only cows in their first lactation are considered here.

model performance ($p$-value $< 10^{-16}$). In the case of autumn calving (figure 2*a*), the proposed equation fits the entire lactation cycle. For winter calving (figure 2*b*), the beginning and the end of the transhumance season appear to be the most challenging periods to be fitted because of a nonlinear slope. The use of equation (3.3) can be illustrated with the autumn calving, with a $d$-parameter of $-0.08$, which is translated by a loss of 144 kg over 60 days. The modelling of protein and fat content curves are also available (electronic supplementary material, S2 and S3).

Total milk production and milk production during alping are reported for the calving months of September and February (figure 3). For the sake of comparison among months, only cows in their first lactation are considered in this graph, as lactation number and calving month are correlated. Cows calving in autumn produce on average 6033 kg during their first lactation, among which 1320 kg are produced in the alp. By contrast, total milk production turns out to be lower for cows calving in winter (5155 kg during their first lactation), while milk production during alping is increased (1755 kg). The $d$-parameters for the two calving seasons being markedly different ($-0.08$ and $-0.02$ for autumn and winter calving, respectively) indicate that productivity is more impacted by alping when calving occurs in autumn than when occurring in winter.

## 4.2. Effect of influencing factors

The significance of the interaction between the group variable and the $d$-parameter is reported (electronic supplementary material, S1). Hereunder, only factors with at least one calving month having a significant $\Delta d$ (i.e. a significantly different impact of alping between the two contrast groups) are presented. The influence of these factors on protein and fat yield is also computed (electronic supplementary material, S4 and S5).

**Figure 4.** The effect of influencing factors is tested by investigating the difference in productivity between two groups of animals coming from contrasted conditions (first and third tertiles, except for THI where the second and third tertile are chosen). Each factor is here reported in a separate column. At the top of each column, the factor name as well as the contrasted groups are reported; the group with highest milk yield during alping is chosen as the reference group, highlighted in red. In each bar plot, the first bar shows the result for autumn calving, and the second for winter calving. The between-group difference in milk production during alping is displayed in the top panel, the between-group difference in milk production during the whole lactation in the intermediate panel and the change in the d-parameter at the bottom. The $\Delta d$-parameter indicates how the reference group is impacted by alping compared with the other group, with positive values meaning lower negative impact (see equation (3.2)). Significant $\Delta d$-values are plotted in black; while grey indicates non-significance. Environmental factors affect production during alping only, making a comparison of the whole milk production redundant (which is why no graph is present in the intermediate panel of the concerned variables). To facilitate the understanding of this graph, the example of lactation (Lact #) is detailed here, where we refer to cows in their third or higher lactation as third lactation cows: third lactation cows produce 5% more milk during alping than the first lactation cows when calving in autumn and even 20% more when calving in winter (m). When considering the whole lactation, third lactation cows produce 15% more milk than first lactation cows when calving in autumn and 19% when calving in winter (n). Third lactation cows calving in autumn are slightly less negatively impacted by alping (positive $\Delta d$) than first lactation cows; an inverse behaviour is observed for winter calving, although both relationships are not significant (o). THI, temperature humidity index, as averaged over 3 (THI-3d) or 30 (THI-30d) days. Prec sp, precipitation in spring; B-region, biogeographic region; Alt (diff), (difference in) altitude; Lact #, lactation number; Preg, pregnancy stage; Height, height at withers; Ft ang: foot angle.

Among environmental conditions, THI, spring precipitation, biogeography and altitude turned out to show a significant effect on milk production during alping (figure 4a–l). Particularly, precipitation in spring and the biogeographic region showed the most important difference on milk production during alping, followed by altitude and altitude difference. Further, calving period appears to interact with environmental conditions, with bigger differences between groups being present in autumn.

The effect of environmental factors is small compared with those of physiological factors, where the biggest effect is found for pregnancy stage for winter calving with a difference in milk production during alping of 20%. Although third and higher lactation cows produce more milk during the whole lactation cycle including alping (figure 4m), they also appear to be more impacted by alping than the first lactation cows as highlighted by negative $\Delta d$-values (figure 4o). The influence of pregnancy stage appears to affect milk production during alping, especially for cows calving in autumn (figure 4p,r). Further, higher cows and/or with steeper foot angle produce more milk both before and during alping than lower ones with gentle foot angle (figure 4s,t,v,w). However, alping appears to negatively impact such cows, especially higher ones (figure 4u,x).

# 5. Discussion

## 5.1. The importance of calving season

The proposed model succeeded in quantifying the impact of alping on milk production by assuming a Wilmink pattern for non-alped cows (figure 2; [40]). This assumption is consistent with literature findings on the same breed [45] and was further validated with animals in our dataset that were alped at a very late stage in their lactation cycle. However, future studies with direct comparisons of lactation curves of alped versus non-alped cows could further corroborate this conclusion. As expected, total milk production resulted globally higher for cows with alping occurring at the end of the lactation, since the drop in production happens later in the cycle. Anyway, winter calving might still be financially attractive for farmers since milk produced in the alps will have a higher economic value on the market and productivity will be higher during alping (figure 3).

Calving season also influences the way an animal is prompt to respond to environmental stress, with a greater impact of transhumance (i.e. greater $d$-parameter in absolute value) for cows calving in autumn and, therefore, alping at the end of their lactation cycle. Increased feed intake is known to have distinct effects on milk production depending on the lactation stage [46], and from what we observe it appears that milk production at the end of the lactation cycle is more sensitive to environmental changes. Similarly, when studying the effect of the considered factors, we showed that the between-group difference in milk production during alping is almost always greater for autumn calving.

## 5.2. Effect of the environment and climate change

Climate change requires species to adapt quickly to new and extreme climatic conditions [47]. In this context, cattle survival and annexed services for humans are threatened because of the low adaptive potential observed for international transboundary breeds [48]. Holstein Fresian cattle, for example, have been shown to be quite sensitive to heat, particularly with THI values above 65 [19]. In Switzerland, climatic conditions are becoming hotter and dryer [49], which exhorts to better understand the effects of climate on cattle welfare and production both at lowland farms and during transhumance. Here, we observe a sensible negative effect of precipitation in spring (figure 4e,f), probably because of its influence on forage growth [31]. Interestingly, heat waves (which are known to highly affect cattle productivity [50]) were found to have minimal impact on milk production during alping, probably because temperatures at high altitude rarely reach problematic thresholds. To further test this hypothesis, several thresholds were tested with values spanning from 63 up to 75: the impact of higher THI remains low (always inferior to 3%), but values obtained from high thresholds should be taken with care, as too few observations are found in these ranges. Similarly, cold spells seem to have an almost negligible influence (electronic supplementary material, S1). The observed effect of biogeographic regions on production can be explained by the difference in spring precipitation between such regions (158 versus 98 mm month$^{-1}$ for the northern flank and the eastern part, respectively). Altitude confirmed its effect on productivity [23], being intrinsically connected with climatic conditions and vegetation type.

## 5.3. Effect of physiological and morphological factors

Lactation number has long been known to strongly influence milk production [51], and this also holds for milk production during alping (figure 4m–o). Even more important, pregnancy stage was found to have a significant impact on milk production during alping, especially when calving occurs in autumn (figure 4p–r). In order to optimize milk yield, cows are generally inseminated a few months after calving, to reach a time span of 1 year between lactation cycles, implying pregnancy stage not to be considered in lactation models to avoid strong collinearity with calving season [15,16]. However, correlation among these variables was not extreme in the present case ($r^2 = 0.8$), most likely because of unsuccessful inseminations leading some cows to delay pregnancy. These results must be interpreted with care, as cows with an early pregnancy are prone to fertility problems.

Many recent research efforts focused on increasing yield in cattle, leading to augmented cattle size [52] but disregarding important side effects such as the loss of adaptive traits through genetic erosion [53]. This phenomenon might become deleterious for transhumance. For instance, despite showing higher productive performances even at alping, higher cows appear to be more impacted when moved to high mountain

pastures (figure 4s–u). As for foot angle, steep angle is associated with a smaller risk of developing hoof diseases [54]. Cows with steeper foot angle were observed to produce more milk both in lowland farm and during alping, but this factor appears to be have limited impact on the $d$-parameter (figure 4v–x).

As further analyses, it would be interesting to determine the impact of the estimated breeding value of the animal, as it is a commonly used measure in the selection of better-performing animals [52]. This would assess how higher ranked animals (i.e. exhibiting better performances under normal conditions) are affected by alping and, therefore, indicate if the current selection is beneficial or damaging to alped cows.

## 5.4. Limitations

Traditionally, lactation modelling is performed on an individual basis, and usually relies on daily or weekly milk records [55]. Here, we based our work on a database composed of monthly milk records, which required the transformation of the data into daily averages over thousands of cows to avoid over-parametrization in the model. This averaging might have diluted the strength of the effect we investigated.

Moreover, the proposed approach still misses validation, which could be achieved by relying on individual observations recorded daily or weekly and belonging to different breeds from the one used here.

Next, the amount of observations among calving months was not constant in the dataset, which possibly made the estimates from the winter months less robust. Further, a hidden age effect—as older cows tend to calve later in the season—could have biased the observed differences in milk production among groups.

Last but not least, the model does not explicitly take into account cow feeding during alping, which is likely to affect milk production [24]. Indeed, the use of concentrate feeding varies among alps and among cows of the same alp. Particularly, differences in milk yield with different calving season could be globally influenced by varying concentrate feeding, with cows at an early stage in the lactation cycle—and thus producing a substantial amount of milk—potentially receiving more concentrates. In a similar context, other studies estimate a herd effect by evaluating the difference among farms, to consider (among other things) different management strategies (see [47,52,56]). In our case, this was not possible, as animals are held in hundreds of farms and are then brought to hundreds of different alps, and no distinction exists to group these farms or alps into two distinct groups as done for other factors, where we compared the first versus the third tertile.

# 6. Conclusion

Transhumance is a traditional farming practice which supports the preservation of both agricultural biodiversity and the socio-cultural heritage of human communities. Nevertheless, a loss in productivity is typically linked with alped livestock, which might discourage farmers from pursuing transhumance and poses its beneficial side effects on ecosystems under threat. Here, we combined biological, geo-environmental and computer science tools to better understand the influence of environmental, physiological and morphological factors on milk productivity during transhumance. We relied on high-resolution meteorological data and five million georeferenced monthly milk records as collected from over 200 000 Braunvieh cows in Switzerland. We show that both environmental and morphological factors have limited influence on animal production, with dry conditions in spring being nevertheless the most affecting environmental factor. This evidence suggests that animal production during transhumance might become even more insecure in future years due to climate change, and stresses, therefore, the urgency of devising strategies to protect this practice. However, the effects of environmental variables are small compared with the ones of physiological factors that have long been known to influence lactation performances (lactation number, pregnancy stage); these factors indeed strongly impact milk production throughout the whole lactation cycle, including during the alping period.

Data accessibility. The data were provided from the Braunvieh-CH association, under the explicit conditions that they will not be shared nor used for other studies. However, a partial dataset is available from the Dryad Digital Repository: https://datadryad.org/stash/dataset/doi:10.5061/dryad.z612jm68g [57], with the average milk production during alping from 20 000 cows, together with lactation information (calving date, lactation number) and environmental data at the location of alping. Cows were chosen randomly, with equal number of animals per year, lactation number and calving month. Furthermore, researchers interested in performing studies on these data may contact directly the association (see contact information homepage.braunvieh.ch). Relevant code for this research work is

stored in GitHub: https://github.com/SolangeD/lactModel and has been archived within the Zenodo repository https://www.doi.org/10.5281/zenodo.3889931.

Authors' contributions. S.D. performed most of the analyses with the help of A.B. C.F. and S.J. supervised the work. S.D. wrote the first draft of the article and all authors contributed in improving this draft.

Competing interests. We have no competing interests.

Funding. No funding source to declare.

Acknowledgements. We are grateful to the breeding organization Braunvieh Schweiz for extracting and distributing the full dataset from their database.

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
