## [Reviewer comments · Royal Society Open Science]

Review History

RSOS-200638.R0 (Original submission)

Review form: Reviewer 1

Is the manuscript scientifically sound in its present form?

Yes

Are the interpretations and conclusions justified by the results?

Yes

Is the language acceptable?

Yes

Do you have any ethical concerns with this paper?

No

Have you any concerns about statistical analyses in this paper?

No

Recommendation?

Accept with minor revision (please list in comments)

Comments to the Author(s)

The paper is of interest as it adapted models of lactation curves to alping conditions, which is still an important production system in mountain areas. Calculations were done on a sufficient dataset from Braunvieh cattle (> 200,000 cows). Animal data were linked to geo-referenced animal data. A limitation is that data from non-alped cows as comparison were not available. It is a well-written paper, given the limitations presented at the end of the discussion.

Some specific comments:

P1, L44: What is meant by adverse conditions?

P3, L55: What was the reason to use these indexes?

P7, L40: What is meant by industrial breeds and which production conditions are implied for these breeds?

P7, L45: Discussion should be extended in problematic thresholds, where are these thresholds for Braunvieh and for 'industrial breeds'?

Review form: Reviewer 2

Is the manuscript scientifically sound in its present form?

Yes

Are the interpretations and conclusions justified by the results?

Yes

Is the language acceptable?

Yes

Do you have any ethical concerns with this paper?

No

Have you any concerns about statistical analyses in this paper?

Yes

Recommendation?

Major revision is needed (please make suggestions in comments)

Comments to the Author(s)

The paper "Big dairy data to unravel effects of environmental, physiological and morphological factors on milk production of mountain-pastured Braunvieh cows" aims to propose a new model of milk production accounting for transhumance. A data base with five million monthly milk records from over 200,000 Braunvieh and Original Braunvieh cows has been used.

Transhumance is a traditional practice in Alpine dairy systems, and it is linked to several ecosystem services. However, milk production can be negatively affected by this practice, and the analysis of the impact of transhumance on milk production can help to define sustainable production practices.

The results are clearly presented and the paper is well written.

The Authors clearly stated the limitations of the study, but in my opinion some of these limitations need to be further considered and discussed.

In particular, it was not possible to compare the milk production of alped cows with no alped cows; does it mean that all the Braunvieh cows are moved to summer farms? I think that a comparison could support the conclusion of this paper.

Considering the relevance of cheese production, I ask to the Authors to explain why they did not consider the evolution of milk quality. Several papers discuss the evolution of milk composition due to transhumance, this dataset could be very useful for this kind of analysis.

In the Lactation curve modelling section (page 6) the Authors explain that they analyzed several animals at a time; this part is not completely clear to me, I ask to the Authors to add some elements to explain the structure of this analysis (how many cows per time, how to consider the measures repeated on the same cows...). As the Authors wrote this approach doesn't allow to validate the model, I think that this point is very relevant for the interpretation of the results.

Lines 38-40 page 6: data from lowland cows were excluded; why those data were not used for a comparison?

Data on management systems were not considered in the analysis. differences due to different level of concentrates, different grazing management, different management in the permanent farms can strongly affect the milk production. Probably it is possible to add the effect of the herd of origin (lowland farms) of the cows and the effect of the summer farm to correct for the different management. I ask to the Authors to further discuss this point.

In the conclusion the implications of this study on the definition of sustainable strategies/policies supporting transhumance practices should be better defined.

Decision letter (RSOS-200638.R0)

Dear Dr Joost

On behalf of the Editors, I am pleased to inform you that your Manuscript RSOS-200638 entitled "Big dairy data to unravel effects of environmental, physiological and morphological factors on milk production of mountain-pastured Braunvieh cows" has been accepted for publication in Royal Society Open Science subject to minor revision in accordance with the referee suggestions. Please find the referees' comments at the end of this email.

The reviewers and handling editors have recommended publication, but also suggest some minor revisions to your manuscript. Therefore, I invite you to respond to the comments and revise your manuscript.

- Ethics statement

- Data accessibility

It is a condition of publication that all supporting data are made available either as supplementary information or preferably in a suitable permanent repository. The data accessibility section should state where the article's supporting data can be accessed. This section should also include details, where possible of where to access other relevant research materials such as statistical tools, protocols, software etc can be accessed. If the data has been deposited in an external repository this section should list the database, accession number and link to the DOI

for all data from the article that has been made publicly available. Data sets that have been deposited in an external repository and have a DOI should also be appropriately cited in the manuscript and included in the reference list.

<http://datadryad.org/submit?journalID=RSOS&manu=RSOS-200638>

- **Competing interests**

- **Authors' contributions**

- **Acknowledgements**

- **Funding statement**

Because the schedule for publication is very tight, it is a condition of publication that you submit the revised version of your manuscript before 10-Jun-2020. Please note that the revision deadline will expire at 00.00am on this date. If you do not think you will be able to meet this date please let me know immediately.

When submitting your revised manuscript, you will be able to respond to the comments made by the referees and upload a file "Response to Referees" in "Section 6 - File Upload". You can use this to document any changes you make to the original manuscript. In order to expedite the

processing of the revised manuscript, please be as specific as possible in your response to the referees. We strongly recommend uploading two versions of your revised manuscript:

If your manuscript is newly submitted and subsequently accepted for publication, you will be asked to pay the article processing charge, unless you request a waiver and this is approved by Royal Society Publishing. You can find out more about the charges at <https://royalsocietypublishing.org/rsos/charges>. Should you have any queries, please contact openscience@royalsociety.org.

on behalf of Professor Marcelo Sanchez (Associate Editor)

Reviewer comments to Author:

Reviewer: 1

Comments to the Author(s)

The paper is of interest as it adapted models of lactation curves to alping conditions, which is still an important production system in mountain areas. Calculations were done on a sufficient dataset from Braunvieh cattle (> 200,000 cows). Animal data were linked to geo-referenced animal data. A limitation is that data from non-alped cows as comparison were not available. It is a well-written paper, given the limitations presented at the end of the discussion.

Some specific comments:

P1, L44: What is meant by adverse conditions?

P3, L55: What was the reason to use these indexes?

P7, L40: What is meant by industrial breeds and which production conditions are implied for these breeds?

P7, L45: Discussion should be extended in problematic thresholds, where are these thresholds for Braunvieh and for 'industrial breeds'?

Reviewer: 2

Comments to the Author(s)

The paper "Big dairy data to unravel effects of environmental, physiological and morphological factors on milk production of mountain-pastured Braunvieh cows" aims to propose a new model of milk production accounting for transhumance. A data base with five million monthly milk records from over 200,000 Braunvieh and Original Braunvieh cows has been used.

Transhumance is a traditional practice in Alpine dairy systems, and it is linked to several ecosystem services. However, milk production can be negatively affected by this practice, and the analysis of the impact of transhumance on milk production can help to define sustainable production practices.

The results are clearly presented and the paper is well written.

The Authors clearly stated the limitations of the study, but in my opinion some of these limitations need to be further considered and discussed.

In particular, it was not possible to compare the milk production of alped cows with no alped cows; does it mean that all the Braunvieh cows are moved to summer farms? I think that a comparison could support the conclusion of this paper.

Considering the relevance of cheese production, I ask to the Authors to explain why they did not consider the evolution of milk quality. Several papers discuss the evolution of milk composition due to transhumance, this dataset could be very useful for this kind of analysis.

In the Lactation curve modelling section (page 6) the Authors explain that they analyzed several animals at a time; this part is not completely clear to me, I ask to the Authors to add some elements to explain the structure of this analysis (how many cows per time, how to consider the measures repeated on the same cows...). As the Authors wrote this approach doesn't allow to validate the model, I think that this point is very relevant for the interpretation of the results.

Lines 38-40 page 6: data from lowland cows were excluded; why those data were not used for a comparison?

Data on management systems were not considered in the analysis. Differences due to different level of concentrates, different grazing management, different management in the permanent farms can strongly affect the milk production. Probably it is possible to add the effect of the herd

of origin (lowland farms) of the cows and the effect of the summer farm to correct for the different management. I ask to the Authors to further discuss this point.

In the conclusion the implications of this study on the definition of sustainable strategies/policies supporting transhumance practices should be better defined.

Author's Response to Decision Letter for (RSOS-200638.R0)

See Appendix A.

Decision letter (RSOS-200638.R1)

Dear Dr Joost,

It is a pleasure to accept your manuscript entitled "Big dairy data to unravel effects of environmental, physiological and morphological factors on milk production of mountain-pastured Braunvieh cows" in its current form for publication in Royal Society Open Science.

on behalf of Professor Marcelo Sanchez (Associate Editor)

Appendix A

Rebuttal letters

Reviewer: 1

We thank the reviewer for his relevant comments. You find the answers to raised questions below:

P1, L44: What is meant by adverse conditions? We changed adverse with “detrimental” in the summary.

P3, L55: What was the reason to use these indexes? We added the following sentence (section Data – climatic data):

“These indices assess all relevant climatic conditions for the evaluation of “hot”/”cold” sensation instead of focusing on temperature only.”

P7, L40: What is meant by industrial breeds and which production conditions are implied for these breeds? The word industrial was indeed not precise enough. We changed it to international transboundary breeds and give a more precise example (section Discussion – Effect of the environment):

“[...] low adaptive potential observed for industrial international transboundary breeds [46]. Holstein Friesian cattle for example has been shown to be quite sensitive to heat, particularly with THI values above 65 [19].”

P7, L45: Discussion should be extended in problematic thresholds, where are these thresholds for Braunvieh and for ‘industrial breeds’? The following sentence was added (section Discussion – Effect of the environment):

“[Interestingly, heat waves (which are known to highly affect cattle productivity [48]) were found to have minimal impact on milk production during alping, probably because temperatures at high altitude rarely reach problematic thresholds.] To further test this hypothesis, several thresholds were tested with values spanning from 63 up to 75: the impact of higher THI remains low (always inferior to 3%), but values obtained from high thresholds should be taken with care as too few observations are found in these ranges. “

Reviewer: 2

We thank the reviewer for his relevant comments. You find the answers to raised questions below:

The Authors clearly stated the limitations of the study, but in my opinion some of these limitations need to be further considered and discussed.

In particular, it was not possible to compare the milk production of alped cows with no alped cows; does it mean that all the Braunvieh cows are moved to summer farms? I think that a comparison could support the conclusion of this paper. In fact only around ¼ of all Braunvieh cows are alped. While the comparison with non-alped cows could have been informative, we did not have access to the data from non-alped cows for this study. However, lactation curves of Braunvieh cows have been shown to follow a Wilmink pattern and cows (see below) that are alped late in the season follow this pattern until almost the end of the lactation cycle. We added this information to the manuscript and modified the following sentences (section Data):

“Importantly, a direct comparison with non-alped cows was not possible because we did not have access to these data [...].”

We also added (beginning of section Discussion):

“The proposed model succeeded in quantifying the impact of alping on milk production by assuming a Wilmink pattern for non-alped cows. This assumption is consistent with literature findings on the same breeds [45] and was further validated with animals in our dataset that were alped at a very late stage in their lactation cycle. However, future studies with direct comparisons of lactation curves of alped versus non-alped cows could further corroborate this conclusion.”

Considering the relevance of cheese production, I ask to the Authors to explain why they did not consider the evolution of milk quality. Several papers discuss the evolution of milk composition due to transhumance, this dataset could be very useful for this kind of analysis.

These analyses were in fact performed, but we noticed that the thread of the article was difficult to maintain with these additional results. However, we acknowledge the utility of investigating e.g. protein and fat yield and percentage and added them in supplementary materials without further discussion in the main document (Sup. Mat. 2-5). Explanation is given (section Data – Milk records and animal information):

“To keep the reader focused on the main thread of the article, our study specifically analyses milk production in terms of quantity (milk yield); however results from computations with protein and fat content and yield are also available in supplementary materials (Sup. Mat. S2-S5).”

References to these figures are also made in the result section where appropriate.

In the Lactation curve modelling section (page 6) the Authors explain that they analyzed several animals at a time; this part is not completely clear to me, I ask to the Authors to add some elements to explain the structure of this analysis (how many cows per time, how to consider the measures repeated on the same cows...). As the Authors wrote this approach doesn't allow to validate the model, I think that this point is very relevant for the interpretation of the results. For a given test-day, all cows from the same calving month having a record on that day were included in this average. We changed the sentence to clarify this (section Methods – lactation curve modelling):

“Therefore, we analysed averaged values by computing, for each test-day, the mean of all available records of that particular Day In Milk (DIM, or number of days after calving). Given that records from the same animal are one month apart but that they are not taken on the same DIM for all animals, the average of milk records for each test-day will constitute a smooth curve with daily values (as displayed in point observations of Fig. 2) [...] Moreover, cows were grouped according to their calving month.”

Lines 38-40 page 6: data from lowland cows were excluded; why those data were not used for a comparison? All cows in our data are in fact alped. However some cows are alped at a later date, so that, during the alping season, we had to remove them from the average to represent only alped animals during this time frame. To clarify this situation, we changed the sentence accordingly (section Methods – lactation curve modelling)

“As dates at which cows are alped or brought back to the lowland farm slightly differ among animals, records from cows remaining at the lowland farm during the alping season (between the 15th of May and the 31st of August) were excluded from this average computation [...]”.

Furthermore, as explained previously, cows that are alped very late in the lactation cycle are used to validate the hypothesis of a Wilmink pattern until the quasi end of the lactation (section Discussion):

“This assumption [Wilmink pattern for non-alped cows] was further validated with animals in our dataset that were alped at a very late stage in their lactation cycle.”

Data on management systems were not considered in the analysis. Differences due to different level of concentrates, different grazing management, different management in the permanent farms can strongly affect the milk production. Probably it is possible to add the effect of the herd of origin (lowland farms) of the cows and the effect of the summer farm to correct for the different management. I ask to the Authors to further discuss this point. Most studies in a similar context are indeed designed to take a herd effect into account. However, in our study, this would only have been feasible with a low number of farms, which is clearly not the case since animals are held in hundreds of farms and alps. Indeed, the way we investigate effects of influencing factors, is by creating two contrast groups. Here, no distinction exists to group these (hundreds of) farms or alps into two distinct groups as done for other factors. Please remember that since we work with averaged values for each test-day, we cannot simply add a random effect as usually done in this context. Last but not least, some alps are known to adopt different management strategies among animals of the same alps, depending on their owner or lactation stage. We added following sentences (end of section Discussion):

“In a similar context, other studies estimate a herd effect by evaluating the difference among farms, to consider (among other) different management strategies (see for example [47,52,56]). In our case, this was not possible, as animals are held in hundreds of farms and are then brought to hundreds of different alps, and no distinction exists to group these farms or alps into two distinct groups as done for other factors, where we compared the first versus the third tertile.”

In the conclusion the implications of this study on the definition of sustainable strategies/policies supporting transhumance practices should be better defined. We changed the sentence in the abstract that talked about sustainable strategies, which was indeed misleading :

“The present findings help to anticipate the effect of climate change and to identify problematic environmental conditions by comparing their impact with factors that are known to influence lactation curves”.